# Accurate and efficient estimation of local heritability using summary statistics and the linkage disequilibrium matrix

Hui Li[1], Rahul Mazumder[2] & Xihong Lin [1,3] ✉

Existing SNP-heritability estimators that leverage summary statistics from genome-wide association studies (GWAS) are much less efficient (i.e., have larger standard errors) than the restricted maximum likelihood (REML) estimators which require access to individual-level data. We introduce a new method for local heritability estimation—Heritability Estimation with high Efficiency using LD and association Summary Statistics (HEELS)—that significantly improves the statistical efficiency of summary-statistics-based heritability estimator and attains comparable statistical efficiency as REML (with a relative statistical efficiency >92%). Moreover, we propose representing the empirical LD matrix as the sum of a low-rank matrix and a banded matrix. We show that this way of modeling the LD can not only reduce the storage and memory cost, but also improve the computational efficiency of heritability estimation. We demonstrate the statistical efficiency of HEELS and the advantages of our proposed LD approximation strategies both in simulations and through empirical analyses of the UK Biobank data.

In the last decade, advances in biotechnology have enabled the estimation of genetic variance contributed by genotyped variants without requiring assumptions about the shared environmental effects[1–6]. These methods estimate the so-called SNP-heritability ($h^2_{SNP}$), defined as the proportion of phenotypic variance caused or tagged by genotyped variants, and have led to critical insights into the genetic architectures of complex traits and diseases[7]. The existing $h^2_{SNP}$-estimation methods can be broadly categorized as either based on individual-level genotypic and phenotypic data[1,2,8–11], or based on summary statistics from genome-wide association studies (GWAS)[3,4,6,12].

Estimators that use individual-level data are generally more precise (i.e., have smaller standard errors), but the applicability of these methods is limited due to data-sharing restrictions. On the other hand, $h^2_{SNP}$-estimation methods that are based on GWAS summary statistics are more widely applicable, but they suffer the drawback of low statistical efficiency[11,13]. For instance, studies have shown that the variance of the $h^2_{SNP}$ estimates from LD-score regression (LDSC) is much larger than that of a REML-based estimator[14,15]. Other methods, such as Generalized Random Effects (GRE) and Randomized HE-regression

(RHE-reg) are also less precise than REML based on simulation results[10,11]. To address the limitations of existing methods, we introduce Heritability Estimation with high Efficiency using LD and association Summary Statistics (HEELS)—an accurate and statistically efficient estimator of $h^2_{SNP}$ that only requires summary-level statistics. Our method is applicable and most suitable to local heritability estimation.

Our work has two main contributions to the literature of $h^2_{SNP}$ estimation. First, we propose an iterative procedure that uses variant-level statistics to solve the REML-score equations by transforming the Henderson's algorithm for variance component estimation in linear mixed models (LMMs)[16,17]. The variant-level statistics required by HEELS includes the marginal association statistics from GWAS and the correlation statistics estimated in-sample (i.e., empirical LD matrix). We show both analytically and through extensive simulations that the HEELS estimator attains a comparable level of statistical efficiency or precision as the conventional individual-level-data-based REML estimators, such as GREML[1] and BOLT-REML[2]. The relative efficiency of HEELS is significantly higher

---

[1]Harvard T.H. Chan School of Public Health, Department of Biostatistics, Boston, MA, USA. [2]Massachusetts Institute of Technology, Operations Research and Statistics group, Cambridge, MA, USA. [3]Harvard University, Department of Statistics, Cambridge, MA, USA. ✉e-mail: xlin@hsph.harvard.edu

than that of the state-of-the-art summary-statistics-based methods, i.e., the estimates from HEELS have much smaller variances.

Second, we introduce a low-dimensional representation of the LD as the sum of a banded and a low-rank matrix ("Banded + LR"). This structure is motivated by the fact that releasing the empirical LD is beneficial for many downstream analyses that utilize variant-level statistics[10,18], but doing so is difficult due to the high dimensionality of LD. Although efforts to release large LD matrices are underway[19], better approximation strategies are needed to facilitate efficient sharing of the LD matrix and to improve the computational efficiency of analyses that involve LD. In this work, we show that our proposed "Banded + LR" structure can ameliorate the challenges associated with LD in heritability estimation. We use an optimization approach to solve for the best representation of LD and we evaluate the performance of different low-dimensional approximations in the context of $h_{SNP}^2$ estimation. Compared to existing methods, our proposed approach not only achieves greater approximation accuracy, but also yields heritability estimates that are less biased.

We applied HEELS to estimate the local SNP-heritability of a wide range of complex traits and diseases in the UK Biobank (UKB). Consistent with our simulation results, the estimates from HEELS are highly concordant with the estimates from REML. We used the more precisely estimated local $\hat{h}_{SNP}^2$ estimates from HEELS to contrast the genetic architectures and to prioritize genetic regions that are enriched for related traits (e.g., lipid traits, blood-cell and leukocyte traits).

In summary, we focused on improving the statistical efficiency of the heritability estimator using variant-level summary statistics. We chose LMM as the simplest and most widely understood model to demonstrate the merits of our approach in terms of statistical efficiency gain both conceptually and empirically. We hope our work can encourage future development of methods that produce more precise heritability estimates.

## Results

### Overview of HEELS

In a nutshell, HEELS uses variant-level statistics, including marginal association statistics and the LD matrix, to estimate SNP-heritability by iteratively solving the REML score equation[17]. Because the score equations solved by HEELS are identical to those solved by GREML[1], HEELS yields heritability estimates that are as precise as those based on REML-based estimates which typically require individual-level data. We first introduce some notations and define $h_{SNP}^2$ under the linear mixed model, assuming individual-level data is available.

Let $\mathbf{y}$ be a length-$n$ vector that denotes the phenotypes of $n$ samples. Denote by $\mathbf{X} \in \mathbb{R}^{n \times p}$ the genotype matrix of $n$ individuals based on $p$ markers or SNPs. We standardize $\mathbf{X}$ and $\mathbf{y}$ such that the variance of the phenotypes is 1 and the variance of each marker-specific genotype vector is $1/p$, or $diag(\mathbf{X}^\top\mathbf{X}/n) = 1/p$. We use an additive genetic model for the phenotypes as $\mathbf{y} = \mathbf{X}\boldsymbol{\beta} + \boldsymbol{\epsilon}$, where $\boldsymbol{\beta}$ is a $p \times 1$ vector assumed to follow $N(0, \sigma_g^2\mathbf{I}_p)$ and $\boldsymbol{\epsilon}$ is a length-$n$ vector distributed as $\boldsymbol{\epsilon} \sim N(0, \sigma_e^2\mathbf{I}_n)$. Under these assumptions, $\mathbf{y} \sim N(0, \mathbf{V})$, where the variance-covariance matrix is $\mathbf{V} \equiv var(\mathbf{y}) = \sigma_g^2\mathbf{X}\mathbf{X}^\top + \sigma_e^2\mathbf{I}_n$. SNP-heritability is defined as $h_{SNP}^2 = \sigma_g^2/(\sigma_g^2 + \sigma_e^2)$ (Methods).

Now suppose we do not have access to the individual-level data, $\mathbf{X}$ and $\mathbf{y}$, and are only provided with the marginal association statistics, $\mathbf{S} = \mathbf{X}^\top\mathbf{y}$ and the LD matrix $\mathbf{R} = \mathbf{X}^\top\mathbf{X}$. (For the sake of simplicity of exposition, we omit the scaling by $1/\sqrt{n}$ for $\mathbf{S}$ or $1/n$ for $\mathbf{R}$, as the scaling does not affect the derivation of the HEELS estimator.) We show that the REML score equations for $(\sigma_g^2, \sigma_e^2)$ can be solved using these summary-level statistics, i.e., $\mathbf{S}$ and $\mathbf{R}$ only, by applying the Sherman-Woodbury matrix identity to the Anderson's algorithm for solving the variance components (Supplementary Notes). HEELS iterates between updating the Best Linear Unbiased Predictor (BLUP) estimates of the joint effect sizes $\widehat{\boldsymbol{\beta}}$ and updating the variance component estimates $(\sigma_g^2, \sigma_e^2)$ until convergence. Let the superscript $^{(t)}$ denote the value of a

variable or a parameter at iteration $t$. The HEELS estimation procedure is as follows:

1. Update the BLUP joint effect size estimates using:

$$\widehat{\boldsymbol{\beta}}^{(t)} = [\mathbf{W}^{(t)}]^{-1}\mathbf{S}, \text{ where } \mathbf{W}^{(t)} = \frac{[\widehat{\sigma_e}^2]^{(t)}}{[\widehat{\sigma_g}^2]^{(t)}}\mathbf{I}_p + \mathbf{R} \qquad (1)$$

2. Update $\sigma_g^2$ using:

$$[\widehat{\sigma_g}^2]^{(t+1)} = \frac{\widehat{\boldsymbol{\beta}}^{(t)\top}\widehat{\boldsymbol{\beta}}^{(t)}}{p - tr([\mathbf{W}^{(t)}]^{-1})} \qquad (2)$$

3. Update $\sigma_e^2$ using:

$$[\widehat{\sigma_e}^2]^{(t+1)} = \frac{\mathbf{y}^\top\mathbf{y} - \mathbf{S}^\top\widehat{\boldsymbol{\beta}}^{(t)}}{n} \qquad (3)$$

We initialize $[\sigma_e^2]^{(0)}, [\sigma_g^2]^{(0)}$ with some random values on $(0, 1)$, and repeat steps 1–3 until convergence, i.e., change in $\hat{\sigma}_g^2, \hat{\sigma}_e^2$ between two consecutive iterations is sufficiently small. The HEELS estimator for heritability is $\hat{h}_{HEELS}^2 = \hat{\sigma}_g^2/(\hat{\sigma}_g^2 + \hat{\sigma}_e^2)$.

We provide derivation details on the updating Eqs. (1–3) and further explain the intuition behind these expressions in the Supplementary Notes. Notably, the algorithm above does not use the raw individual-level data $\mathbf{X}$ or $\mathbf{y}$, but only utilizes summary statistics $\mathbf{S}$ and $\mathbf{R}$, which are sufficient statistics for the variance component parameters in our model. When sample variance $\mathbf{y}^\top\mathbf{y}/n$ is known, implementing the procedure above is straightforward. When it is not known, we can use the Z-scores, approximate $\mathbf{y}^\top\mathbf{y}/n$ by 1 and rescale $\hat{\sigma}_g^2, \hat{\sigma}_e^2$ at each iteration. Similar approximation strategies have been adopted by other summary-statistics-based methods[20].

The primary reason for the high statistical efficiency of HEELS is that the estimating equations of $(\sigma_g^2, \sigma_e^2)$ solved by HEELS are identical to the REML score equations of $(\sigma_g^2, \sigma_e^2)$[17]. We developed our procedure by applying the Woodbury matrix identities to Henderson's iterative algorithm[16,17,21], which are known to be useful for computing the maximum-likelihood estimators in a linear mixed model[22,23]. The main contribution of our method lies in the transformation of this algorithm, such that only variant-level statistics are needed. The asymptotic variance of the variance components can be derived using the likelihood theory. Importantly, the variance estimator can also be re-expressed using summary statistics $\mathbf{S}, \mathbf{R}$ only, and thus the uncertainty in our estimates can be quantified without access to the individual-level data as well. We apply the multivariate delta method to obtain a plug-in estimator of the variance of $\hat{h}_{HEELS}^2$ (Methods).

### Accuracy and statistical efficiency of the HEELS estimator

To evaluate the performance of the HEELS heritability estimator, we performed simulations using the genotype array data of $332,430$ unrelated British white individuals in the UK Biobank. We define relative efficiency (RE) of a heritability estimator as the ratio of the variance of the GREML estimator and the variance of the estimator in comparison[24,25]. A high RE implies that the estimator is more precise or has a lower variance. We selected GREML[1], LDSC[3], GRE[10] and HESS[12] as the representative $h_{SNP}^2$-estimators to be compared with HEELS. The other methods are not included in our comparison due to our focus on improving the statistical efficiency of summary-statistics-based heritability estimators. In the Supplementary Notes, we provide a more comprehensive review of the existing $h_{SNP}^2$-estimation approaches, with their key advantages and limitations summarized (Supplementary Table 1).

We first simulated quantitative phenotypes using the LMM model, where the causal effect sizes of all genetic markers follow a normal

distribution and contribute to the genetic variance equally. We found that HEELS is unbiased in finite samples and attains high statistical efficiency. In particular, we found a high degree of concordance between the estimates from REML-based estimators (such as GREML and BOLT-REML) which use individual-level data and those from HEELS which only relies on variant-level summary statistics (Fig. 1). When the full set of samples in the UKB is used, the relative statistical (RE) efficiency of HEELS reaches as high as 99.44%, whereas GRE and LDSC have much lower RE's of 19.81% and 8.68%, respectively (Table 1).

LDSC's lack of statistical efficiency has been observed in previous studies[12,14,26,27]. We provide a stylized argument to contrast the statistical efficiency of LDSC and HEELS under the framework of moment-matching methods[14] (Supplementary Notes). Briefly, the score equations solved by HEELS coincide with the estimating equations of the most efficient generalized method-of-moment estimator. Since LDSC can be viewed as a method-of-moment estimator with sub-optimal weights, its statistical efficiency is expected to be lower than that of HEELS. In simulations, we found that GRE and HEELS are both unbiased regardless of sample size, but GRE is less statistically efficient that HEELS (Supplementary Fig. 1). The variability of the GRE estimates is particularly high when sample size is small, as expected from the theory and the derivation of GRE[10]. Even when the full UKB is used, the variance of $\hat{h}^2_{GRE}$ is still five times as large as that of the HEELS estimator.

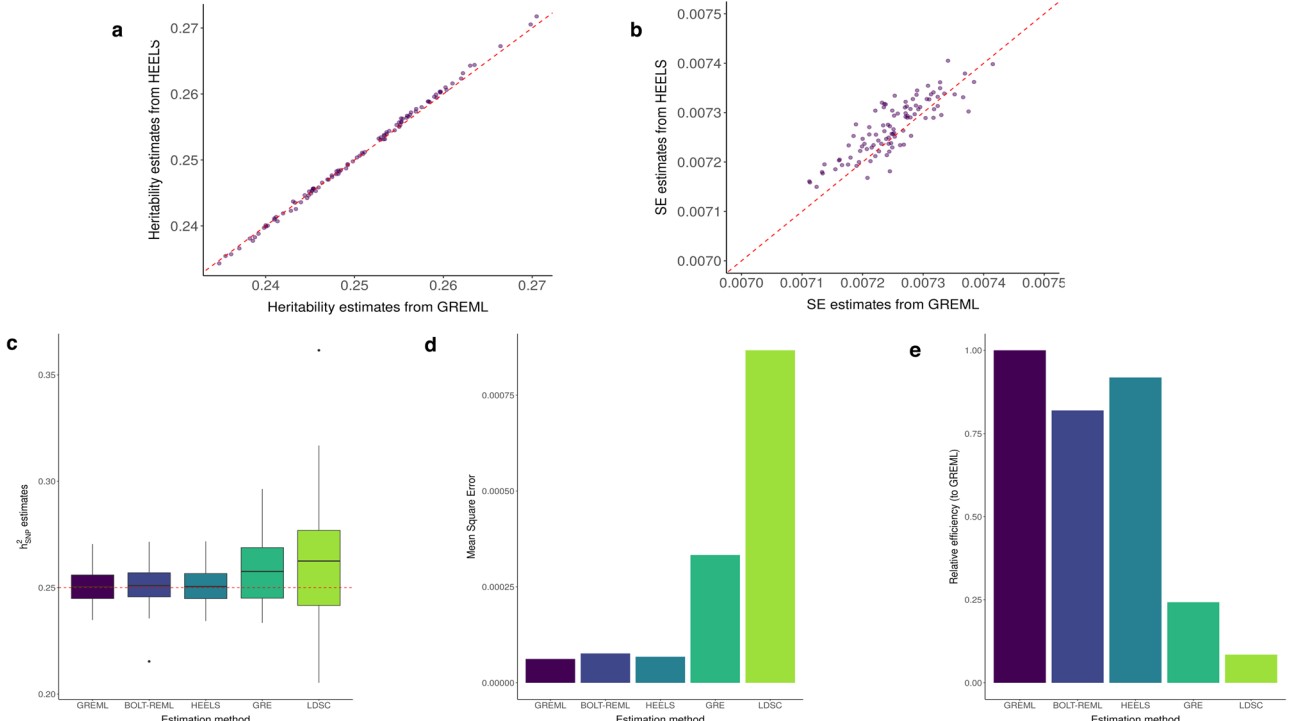

**Fig. 1 | Comparison of the performance of HEELS with different methods using simulation studies.** Simulated phenotypes using real genotypic data from the UK Biobank, array SNPs on chromosome 22 with MAF > 0.01 ($n = 30,000, p = 9,205$). **a** Local $h^2_{SNP}$ estimates from HEELS using summary statistics vs GREML using individual-level data. Red-dotted line: $y = x$. **b** Analytical SE estimates from HEELS vs GREML. Red-dotted line: $y = x$. **c** Distribution of the $h^2_{SNP}$ estimates from 100 simulations using different methods: GREML and BOLT-REML use individual level data; HEELS, GRE and LDSC use summary statistics. Red-dotted line: true $h^2_{SNP}$ of 0.25. The lower and upper hinges correspond to the first and third quartiles (the 25th and 75th percentiles). The upper (lower) whisker extends from the hinge to the largest (smallest) value no further than 1.5 × IQR from the hinge (where IQR is the interquartile range). Data beyond the end of the whiskers are called "outlying" points and are plotted individually. **d** MSEs of the $h^2_{SNP}$ estimates using different methods. **e** Relative efficiency of different methods compared to GREML.

**Table 1 | Relative efficiencies of different summary-statistics based $h^2_{SNP}$ estimators (HEELS, GRE, HESS) compared with individual-level data based estimator (REML) in simulation studies**

| Polygenicity | Sample size | Empirical variance of heritability estimates | | | | Relative efficiency (to REML) | | |
|---|---|---|---|---|---|---|---|---|
| | | REML | HEELS | GRE | LDSC | HEELS | GRE | LDSC |
| 1 | 332k | 1.43E-05 | 1.44E-05 | 7.24E-05 | 1.65E-04 | 99.44% | 19.81% | 8.68% |
| 1 | 140k | 2.15E-05 | 2.20E-05 | 6.53E-05 | 1.77E-04 | 97.82% | 32.94% | 12.17% |
| 1 | 40k | 4.05E-05 | 4.07E-05 | 1.32E-04 | 5.45E-04 | 99.54% | 30.76% | 7.43% |
| 1 | 30k | 6.19E-05 | 6.74E-05 | 2.55E-04 | 7.32E-04 | 91.91% | 24.25% | 8.46% |
| 0.5 | 30k | 6.80E-05 | 7.26E-05 | 2.58E-04 | 8.71E-04 | 93.69% | 26.38% | 7.81% |
| 0.2 | 30k | 9.35E-05 | 1.00E-04 | 2.98E-04 | 8.89E-04 | 93.16% | 31.42% | 10.52% |
| 0.1 | 30k | 1.16E-04 | 1.29E-04 | 4.39E-04 | 8.91E-04 | 90.17% | 26.41% | 13.02% |

Sparsity and sample sizes are varied across settings. REML estimates are computed using GCTA for small sample (30k), and BOLT-LMM for large samples (40k, 140k, 332k). Statistical relative efficiency: ratio between the variance of REML and the variance of an estimator. Polygenicity: the proportion of genetic markers that are causal.

We also compared the performance of HEELS with that of HESS, which is a state-of-the-art method for local heritability estimation[12]. We found that the HESS estimator is sensitive to the degree of LD regularization applied through the truncated SVD (Supplementary Fig. 2), a phenomenon that has been reported in the original paper. In contrast, HEELS is unbiased when the full LD matrix is used, and we explicitly minimize the bias of the heritability estimator when the LD is approximated (see below). It is worth noting that the analytic form of the HESS estimator closely resembles that of the GRE estimator, despite their disparate modeling frameworks. Both GRE and HESS can be viewed as principal-component regression (PCR)-based estimators (Supplementary Notes). In simulations, we indeed observed that the HESS estimates are closer to the GRE estimates as the amount of LD regularization decreases (Supplementary Fig. 2), matching the theory which implies an alignment of $\hat{h}^2_{HESS}$ and $\hat{h}^2_{GRE}$ when LD is estimated in-sample and without regularization.

Next, we considered relaxing the infinitesimal assumption and altered the sparsity of the causal effects. Although this violates the LMM assumption, we render it an important scenario to consider in order to assess the robustness of our estimator. In simulations, we observed that HEELS remains unbiased and statistically efficient under these mis-specified model scenarios (Table 1). We note that sparsity of the causal effects generally increases the variability of the heritability estimates, but the statistical efficiency of HEELS remains the highest among summary-statistics-based methods (Supplementary Fig. 3). Across settings of various levels of polygenicity and sample sizes, HEELS leads to an increase in precision that is equivalent to increasing the GWAS sample size at least 3 times or 7 times, compared to GRE and LDSC, respectively (Supplementary Fig. 4). These results, demonstrating the robustness of the HEELS estimator, are consistent with previous work that showed REML's robustness under sparse genetic architectures, further illustrating the likeness between HEELS and REML. Our findings also corroborate the theoretical findings that suggest REML's consistency under non-infinitesimal models[28] and its larger asymptotic variance under sparse architectures[29].

Finally, we evaluated the Type I error rates controlled by different methods. We found that the standard error of HEELS is well-calibrated regardless of the sample size as long as the model is correctly specified (Supplementary Fig. 5). The standard errors reported by BOLT-REML using individual-level data generally lead to correct coverage but can be anti-conservative when sample size is small. The standard errors from GRE can result in under-coverage, as has been reported (see Supplementary Table 4 of Hou et al.[10]). When the true effect size does not follow a normal distribution, we observe under-coverage for all LMM-based estimators. Nevertheless, the calibration of HEELS is still better than GRE and LDSC, and is most comparable to REML across settings (Supplementary Fig. 6).

## A unified framework to compare LD approximations

It is a well-known challenge in statistical genetics that the LD matrix is expensive to store and compute with. While several low-dimensional representations of the LD have been proposed and used in the literature (e.g., banding[30–32], shrinkage regularization[30,33], truncated SVD[12,34] and low-rank approximation[26,27]), the impact of LD approximations on heritability estimation remains unclear. We aim to construct a low-dimensional representation of the LD matrix that can help reduce the computational cost of HEELS heritability estimation without incurring much loss of accuracy or efficiency in the $h^2_{SNP}$ estimates. To this end, we propose representing the in-sample LD matrix, $\mathbf{R}$, as the sum of a banded matrix and a low-rank matrix ("Banded + LR"),

$$\mathbf{R} \approx \mathbf{R}_b + \mathbf{R}_r = \mathbf{R}_b + \sum_{k=1}^{r} \lambda_k \mathbf{u}_k \mathbf{u}_k^\top = \mathbf{R}_b + \mathbf{U}_r \Lambda_r \mathbf{U}_r^\top, \quad (4)$$

where $\mathbf{R}_b$ is a banded matrix with a central bandwidth of $b$; $\mathbf{R}_r$ is a low-rank matrix with rank $r$; $\lambda_k$ and $\mathbf{u}_k$ (also diagonal entries of $\Lambda_r$ and columns of $\mathbf{U}_r$) are the $k$-th eigenvalues and eigenvectors of $\mathbf{R}_r$. We consider six different strategies to decompose or represent the empirical LD using this "Banded + LR" structure (Table 2), all of which take the form of the expression in Eq. (4), but differ in terms of (1) whether the banded component is a diagonal matrix, in which case the approximation becomes a spiked covariance matrix[35], and (2) whether the banded and low-rank components are constrained to be positive semi-definite (PSD).

The "Banded + LR" approximation provides a unified framework for analyzing and evaluating the performance of various LD approximations, because most of the existing LD approximation strategies can be viewed as special cases of Eq. (4). For instance, regularizing the LD via the truncated SVD is equivalent to using "Banded + LR" while setting $b = 0$; the most common way of approximating the LD is banding, which corresponds to a "Banded + LR" structure with $r = 0$. The motivation behind our proposed "Banded + LR" structure ($b > 0, r > 0$) is twofold. On one hand, since correlations between two genetic markers are typically induced by physical proximity, the vast majority of the non-zero elements of LD matrix lie on the central band. On the other hand, we want to retain the in-sample correlation structure outside of the central band. We found that oftentimes there are appreciable non-zero off-central-band elements in finite samples, and methods such as HEELS utilizes these values to produce accurate estimates.

To solve for the best representation of the LD matrix, we adopt an optimization approach, minimizing $||\mathbf{R} - \widetilde{\mathbf{R}}||_F^2$, where $\widetilde{\mathbf{R}}$ is the approximation in the form specified in the third column of Table 2. We explain the distinctions between the different LD approximations in Methods and provide further details on their respective estimation procedures in Supplementary Table 2. Briefly, the "Seq_Band_LR" strategy first bands the LD matrix and then performs low-rank decomposition on the residual off-banded matrix; the "PSD_Band_LR"

**Table 2 | Summary of the proposed LD approximation methods**

| Structure | Strategy name | Form of Decomposition | Property of the approximation |
|---|---|---|---|
| Spiked covariance | Spike_LR | $Diag(\sigma^2, ... \sigma^2) + \mathbf{U}_r \Lambda_r \mathbf{U}_r^\top$ | PSD as long as $r << p$ |
| Spiked covariance | Spike_PSD | $Diag(\sigma^2, ... \sigma^2) + \mathbf{L}_r \mathbf{L}_r^\top$ | PSD guaranteed |
| Spiked covariance | Spike_PSD_hetero | $Diag(\sigma_1^2, ... \sigma_p^2) + \mathbf{L}_r \mathbf{L}_r^\top$ | PSD guaranteed |
| Banded + Low-rank | Seq_Band_LR | $\mathbf{R}_b + \mathbf{U}_r \Lambda_r \mathbf{U}_r^\top$ | PSD not guaranteed |
| Banded + Low-rank | PSD_Band_LR | $\mathbf{L}_b \mathbf{L}_b^\top + \mathbf{U}_r \Lambda_r \mathbf{U}_r^\top$ | PSD as long as $r << p$ |
| Banded + Low-rank | Joint_Band_LR | $\mathbf{L}_b \mathbf{L}_b^\top + \mathbf{L}_r \mathbf{L}_r^\top$ | PSD guaranteed |

Six methods are proposed to approximate the LD matrix using either (1) a the spiked covariance matrix (Spike_LR, Spike_PSD, Spike_PSD_hetero) or (2) the sum of a banded and a low-rank matrix (Seq_Band_LR, PSD_Band_LR, and Joint_Band_LR). See Methods and Supplementary Notes for further explanations. PSD: positive semi-definite. $b$: bandwidth of the banded component; $r$: rank of the low-rank component. {$\mathbf{L}, \mathbf{U}, \Lambda$}: Cholesky factor, eigenvector and eigenvalue of the target matrix. Subscript of $\mathbf{L}, \mathbf{U}, \Lambda$ represents the component, $b$ for banded or $r$ for low-rank.

strategy first approximates the banded part of the LD matrix using a PSD matrix and then performs low-rank decomposition on the residual matrix; the "Joint_Band_LR" strategy jointly approximates the banded and the low-rank components using PSD matrices. We have derived the asymptotic variance of our HEELS estimator, which only depends on the variant-level statistics, i.e., the additional variance incurred by the approximation can be quantified without having to access the full LD matrix (Supplementary Notes).

### Efficacy of the banded plus low-rank representation

In simulations, we found that our proposed "Banded + LR" representations approximated the original full LD well and led to more accurate $\hat{h}^2_{HEELS}$ than the existing methods (Fig. 2). We attribute the efficacy of our proposed "Banded + LR" representation to the fact that it retains the central band of the covariance matrix while exploiting the

signals in the residual off-central-band part of the LD matrix. Computationally, the reduction of runtime pertains to our application of the Woodbury formula to the low-rank matrices, which helps circumvent the need of direct matrix inversion in HEELS heritability estimation.

Relative to using the full LD matrix ("Exact_LD"), approximating the LD using a "Banded + LR" structure markedly improves the computational efficiency of HEELS, reducing the runtime by 29.40%, 73.02% and 73.26% for "Seq_Band_LR", "PSD_Band_LR" and "Joint_Band_LR", respectively (Table 3). The greater efficiency gain of the "PSD_Band_LR" and "Joint_Band_LR" strategies is attributable to the PSD assumption (Supplementary Notes). The storage cost of the Banded + LR approximations is also substantially lower (i.e., more than 70%). For fine-tuned approximation settings (e.g., "PSD_Band_LR" with $b = 400$ and $r = 800$), the bias in the heritability estimates is even smaller than that when the full LD is used (relative bias of 0.22% vs

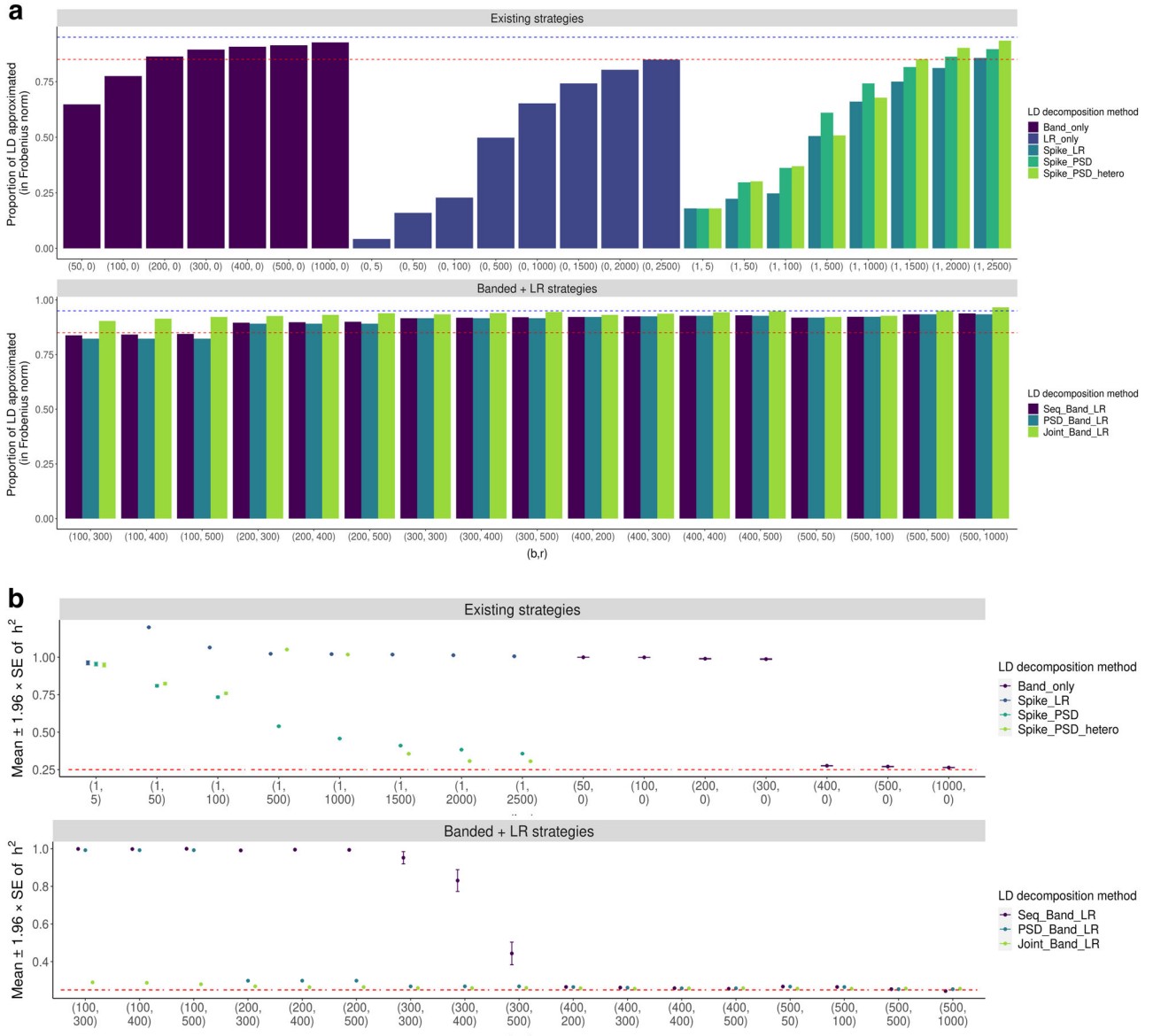

**Fig. 2 | Comparison of the performance of different LD approximation strategies.** Simulated phenotypes using real genotypic data from UK Biobank, array SNPs on chromosome 22 with MAF > 0.01 ($n = 332, 430$, $p = 9, 220$). Labels of the approximation strategies are explained in Table 2. $b$ denotes the bandwidth of the banded component. $r$ denotes the number of factors in the low-rank component. **a** Approximation accuracy, measured by $||\widetilde{\mathbf{R}}||_F / ||\mathbf{R}||_F$, where $\widetilde{\mathbf{R}}$ is the approximated

LD matrix. Dotted lines are the reference levels: red−85%; blue−95%. **b** $h^2_{SNP}$ estimates from 100 simulations. Red reference line represents the true heritability value of 0.25. The upper (lower) whisker extends from the mean to the values $1.96 \times SE$ above (below) the mean. The "LR_only" approximations lead to large bias in $h^2$ estimation, so are omitted from the comparison.

**Table 3 | Summary of the performance of different Banded + LR approximation strategies**

| Strategy | Average across hyperparameter settings | | | | | |
|---|---|---|---|---|---|---|
| | LD approximation accuracy | Heritability estimates | | | Runtime | |
| | | Bias | MSE | | In seconds | % saved |
| Exact_LD | 100% | 0.00112 | 1.35E-05 | | 234.60 | - |
| Seq_Band_LR | 92.62% | 0.02185 | 3.43E-03 | | 165.63 | 29.40% |
| PSD_Band_LR | 92.50% | 0.00392 | 3.74E-05 | | 63.29 | 73.02% |
| Joint_Band_LR | 94.45% | 0.01154 | 1.55E-04 | | 62.74 | 73.26% |
| **Strategy** | **Fine-tuned hyperparameter settings** | | | | | |
| | LD approximation accuracy | Heritability estimates | | | Runtime | |
| | | Bias | MSE | | In seconds | % saved |
| Seq_Band_LR | 93.67% | 0.00001 | 1.74E-05 | | 121.44 | 48.24% |
| PSD_Band_LR | 93.57% | 0.00022 | 1.39E-05 | | 44.04 | 81.23% |
| Joint_Band_LR | 95.70% | 0.00795 | 7.88E-05 | | 44.13 | 81.19% |

Simulated phenotypes using real genotypic data from UK Biobank, array SNPs on chromosome 22 with MAF > 0.01 ($n = 332, 430, p = 9, 220$). The top panel shows the average performance across hyperparameter settings, with $b$ varying from 300 to 600 in increments of 100, and $r$ varying from 300 to 800 in increments of 50. The bottom panel shows the performance of the fine-tuned hyperparameter settings: $(b, r) = (500, 500)$ for "Seq_Band_LR", $(b, r) = (400, 800)$ for "PSD_Band_LR", and $(b, r) = (500, 800)$ for "Joint_Band_LR". The second column shows the LD approximation accuracy, measured in Frobenius norm, as $||\tilde{\mathbf{R}}||_F / ||\mathbf{R}||_F$, where $\tilde{\mathbf{R}}$ is the LD approximation. The operating characteristics of the heritability estimates are based on 100 simulated phenotypes. The runtime reported is for the HEELS heritability estimation (i.e., not including the optimization to solve for the optimal representation). The computational cost of the existing LD approximation methods and newly proposed Banded + LR methods is reported in Supplementary Tables 4 and 5 respectively.

1.12%), while runtime and storage decrease by 68.92% and 74.16% respectively. Overall, we determined that our proposed "Banded + LR" representations can improve the computational efficiency of HEELS across hyperparameter settings without incurring large loss of accuracy and efficiency.

In comparing the performance of the three "Banded + LR" strategies, we found that they have varying degrees of sensitivity with respect to changes in the hyperparameter values $b$ and $r$ (Supplementary Fig. 7). Nevertheless, they converge in their approximating abilities (Supplementary Fig. 8) and lead to small bias when $b$ and $r$ are sufficiently large (Supplementary Fig. 9). The bandwidth of the banded component plays a critical role in determining the magnitude of the bias in $h^2_{HEELS}$ as well as runtime, whereas the number of the low-rank factors fine-tunes the estimator (Supplementary Fig. 10). Among the three "Banded + LR" strategies, we recommend using "PSD_Band_LR" as the default, as it is least sensitive to changes in hyperparameter values ($b$ and $r$) and leads to the smallest accuracy and efficiency loss, though it tends to produce slightly upwardly biased $h^2_{SNP}$ estimates (Supplementary Figs. 7–12).

Finally, we developed a data-adaptive procedure to select the optimal hyperparameter values of $b$ and $r$ using cross-validation ("CV"). We refer to this procedure as "pseudo-validation", as it does not require actual held-out samples but only relies on simulated summary statistics (Methods). We validated the effectiveness of this procedure, by verifying that the optimal hyperparameter values obtained from this algorithm indeed yield accurate $\hat{h}^2_{HEELS}$ estimates (Supplementary Fig. 13). To improve the computational efficiency of our tuning procedure, we implemented our own version of the incremental SVD algorithm[36], and adopted several computational techniques to speed up the approximation and the tuning procedure (Supplementary Notes).

**Precise estimates of local heritability in the UK Biobank**
We applied HEELS to estimate the local SNP-heritability of 30 anthropometric, medical and behavioral traits in the UK Biobank (Methods), using the LD blocks estimated by Berisa and Pickrell[37], which have been widely used to proxy approximately independent loci on the human genome[38–40]. We used Z-statistics for all phenotypes and interpret heritability on the liability scale for binary traits[8,41,42]. In line with our expectation based on the theoretical and simulation results, there is an exceptionally high concordance rate between the local heritability estimates from HEELS and those from REML ($r^2 = 0.98$ on average

across the traits), which is the gold standard or the most efficient estimator but requires individual-level data (Fig. 3). In contrast, the correlations between the GRE or HESS estimates and the REML estimates are weaker ($r^2 = 0.88$).

HEELS yields standard errors that are most comparable to those from REML, which is based on individual-level data, and its relative efficiency is higher than all the other summary-statistics based methods (Supplementary Table 3). We applied our hyperparameter tuning algorithm to obtain the optimal representations of the LD blocks, and found that these LD approximations perform well, yielding local $h^2$ estimates that are concordant with those which are based on the full LD (Supplementary Fig. 15). The "Joint_Band_LR" strategy leads to less accuracy loss compared to the other two strategies ("Seq_Band_LR" and "PSD_Band_LR"), plausibly due to its greater capacity of approximating the LD (Supplementary Fig. 17).

**Applications of HEELS: Contrasting polygenicity of complex traits and diseases; Identifying risk loci with putative pleiotropic effects**
Studies have shown that complex traits and diseases have different degrees of polygenicity[43]. Local heritability estimation provides a useful tool for comparing the polygenic architecture of complex traits, as its distribution pattern across the genome can reflect the degree of which heritability is spread among loci[12,44]. Using local heritability estimated with greater precision from HEELS, we found varying degree of polygenicity across a wide range of traits (Supplementary Fig. 18). For instance, the heritability of behavioral traits and anthropometric traits are more evenly spread out across the genome, whereas those of autoimmune diseases and allergic conditions, lipid traits are more localized and disproportionately dispersed on the genome. Alternatively, we evaluated the polygenicity of complex traits by examining the Pearson correlation coefficients between local heritability and the size of the genomic regions. Indeed, we observed a near-perfect linear relationship between chromosome length and the fraction of $h^2_{SNP}$ explained by the chromosome for highly polygenic traits, such as BMI ($r^2 = 0.988$), WHR ($r^2 = 0.919$), educational attainment ($r^2 = 0.991$) and neuroticism ($r^2 = 0.990$). In contrast, the correlations are much lower for less polygenic traits, such as HDL ($r^2 = 0.295$), LDL ($r^2 = 0.537$) and autoimmune diseases ($r^2 = 0.757$) (Supplementary Fig. 19).

Next, we used the local heritability estimated from HEELS to identify risk loci that are enriched for related traits and thus potentially

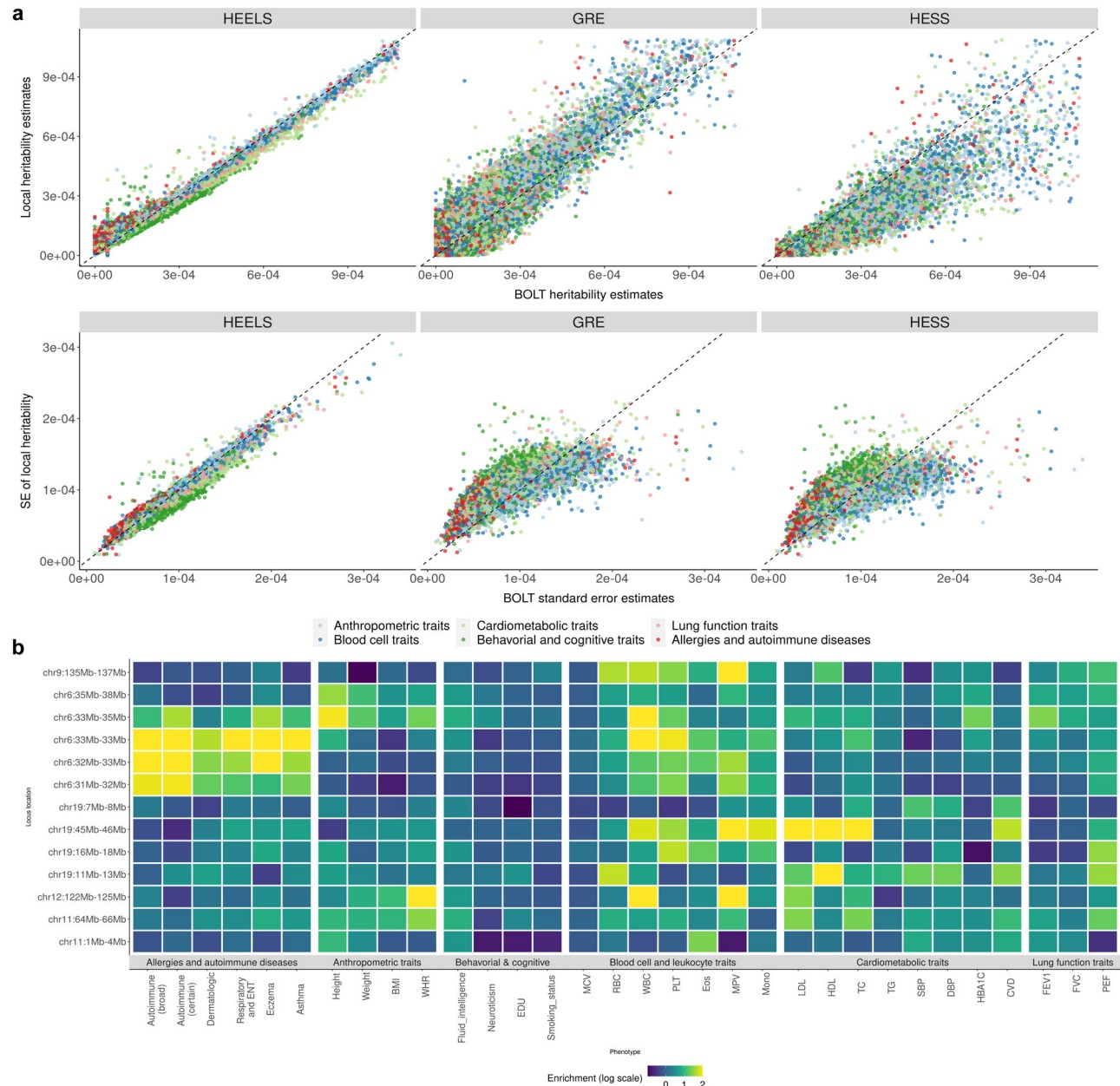

**Fig. 3 | Analysis results from application of HEELS to UKB. a** Comparison of heritability estimates and standard errors between BOLT-REML and summary-statistics-based methods (HEELS, GRE, HESS) (n = 332, 340). Each dot represents a local estimate at one locus for a given trait. Red dashed line: y = x. **b** Heritability enrichment of multiple diseases and traits at putative pleiotropic risk loci. Loci locations on the Y-axis are denoted by the start and end positions (in Mbp). The shade of each box represents the enrichment of heritability for the given region and trait, with log transformation.

host pleiotropic effects (Fig. 3). Our results recapitulate known disease-associated loci from previous findings. For example, we found that the Major Histocompatibility Complex (MHC) (Chr6:31-34Mb) shows strong pleiotropic signals for immunologically relevant diseases and leukocytes involved in innate immunity and inflammatory response. This region, also referred as the human leukocyte antigen (HLA), is known to be highly polygenic[45] and plays a key role in the induction and regulation of immune responses[46–48]. We also identified Chr19:45-46Mb as a heritability hotspot, which is highly enriched for multiple lipid and cardiometabolic traits while being moderately enriched for leukocytes. This locus harbors the gene cluster—*APOE, APOC1, APOC4, APOC2*—which code for apolipoproteins that are known to be responsible for controlling plasma lipid levels, with subsequent implications in cardiovascular pathology[49]. Our results highlight the probable presence of pleiotropic effects regulating lipid metabolism[50,51].

We identified two loci that may provide novel insights into the shared genetic basis between related traits and diseases. First, we found that the Chr19:16-18Mb locus is not only enriched for leukocytes such as platelet, lymphocyte and eosinophil, but also enriched for peak expiratory flow. Recent studies have associated platelet-to-lymphocyte ratio (PLR) with the severity of COVID-19, as the PLR of patients can indicate the degree of cytokine storm[52,53]. Our results suggest a putative link between the inflammatory markers and lung functions on a genetic level, and point to a specific region with putative pleiotropic effects. Further study of this locus may help illustrate its prognostic value for impaired pulmonary functions due to systemic inflammation.

Another region of particular interest is Chr4:143–146 Mb, which is highly enriched for all three of the lung function traits in our analysis (Supplementary Fig. 16). Although no evidence has directly implicated this region for lung functions, the 4q31 locus—which contains this

block—has been prioritized in three previous GWAS studies[54–56] and one meta-analysis[57]. One plausible explanation for the pleiotropic effects of this region is that it hosts the Hedgehog (Hh)-interacting protein (*HHIP*) gene, which is associated with multiple pulmonery traits, such as FEV1/FVC ratio, chronic obstructive pulmonary disease (COPD) and lung cancer[58–61]. As a regulator of the Hh signaling pathway, *HHIP* is vital for embryonic lung development and is also involved in mature airway epithelial repair[62]. We hypothesize that the observed pleiotropic effects at this locus may be attributable to the *HHIP* gene and its broad influence on lung functionality[63,64].

## Discussion

To summarize, we have introduced HEELS, a likelihood-based approach to obtain highly efficient heritability estimators in linear mixed models, using variant-level statistics that can be publicly shared. HEELS requires marginal GWAS summary statistics and the in-sample LD matrix, but yields highly precise estimates that are comparable to REML-based estimators which typically require individual-level data. HEELS outperforms the existing summary statistics-based heritability methods such as LDSC, GRE and HESS, in terms of statistical efficiency.

Another important contribution of our work is that we showcase the benefits of approximating the LD matrix using a banded matrix plus a low-rank matrix. We demonstrate in simulations that this low-dimensional representation can improve the computational efficiency of HEELS heritability estimation but incurs minimal loss of accuracy or efficiency. The pseudo-validation procedure we propose provides a principled way to select hyperparameters for the Banded + LR approximation, in contrast to existing approaches which make somewhat arbitrary assumptions about the structure of the LD. Our proposed low-dimensional approximations of the LD can be employed more generally to facilitate the storage and sharing of large LD matrices.

We discuss several limitations of our current work and outline some future directions. First of all, our estimator is derived based on the LMM, which assumes a normal distribution of the causal effect sizes. This assumption greatly simplifies our comparison between HEELS and REML, and helps illustrate the main conceptual advance we aim to make relating to statistical efficiency. However, these assumptions may be relaxed in two main ways. One is to allow zero effects and incorporate sparse components in modeling the distribution of effect sizes. This direction has been explored in the literature under the framework of Bayesian variable selection, leading to the development of methods, such as BSVR[65], BSLMM[66] and the Bayesian-alphabet models[67]. However, studies have shown that these methods provide less statistically efficient heritability estimators than REML (even when the LMM assumption is met)[68]. Future research is needed to develop heritability estimators that are more efficient than REML under a sparse genetic architecture. Another direction is to adopt a more flexible heritability model, where SNP-specific contributions to the genetic variance varies depending on the allele frequency and the level of LD. This alternative scheme was put forward by authors of the LDAK[9] model and have been examined carefully in their subsequent works. In simulations, we observed that indeed, HEELS can be biased under mis-specified models, and that the amount of bias varies with the strength of MAF/LD dependency (Supplementary Fig. 20). This is not surprising, given the existing evidence in the literature, showing that GREML is biased when SNP-specific genetic variance is not evenly distributed[10,15]. However, we note that HEELS incurs less bias than LDSC and has the lowest MSE across different settings of model mis-specification (Supplementary Fig. 21). Further studies are warranted to extend HEELS and to explicitly accommodate both the sparse effects and the marker-specific weights.

Second, we emphasize that HEELS depends on the in-sample LD. This requirement is related to the fact that we define heritability in the LMM conditional on the genotypes **X**. In simulations, we observe that replacing the empirical LD with the out-of-sample LD in HEELS can lead to biased heritability estimates. Further research on how out-of-sample LD can influence the statistical properties of the HEELS estimator is needed. Currently, we caution against applying HEELS to mismatched LD and GWAS summary statistics. Furthermore, HEELS can potentially be applied to summary statistics from meta-analyses, but the key challenge is how to appropriately integrate the LD from multiple cohorts or sources. One fruitful direction for future research is to explore ways to combine the low-dimensional representations of the LD matrices from different studies[18]. In rare variant association tests, researchers routinely release both the vector of score statistics and the LD matrices when publishing association studies[69–72]. While this is not yet standard practice for common variant analyses, we advocate including the in-sample LD matrix, in potentially approximated form, as part of the summary-level statistics when publishing GWAS results. This will enable researchers to more precisely characterize the genetic architecture of a disease (e.g., through HEELS or GRE), facilitate privacy-preserving benchmarking efforts[73], and alleviate problems related to mismatched ancestry when using external reference panels for meta-analyses[74].

Lastly, we note that the HEELS estimator is best suited for studying and contrasting *local* heritabilities. Applying HEELS to a larger set of markers (e.g., genome-wide estimation) can be difficult for two main reasons. First, the computational cost of running HEELS scales with the number of markers (rather than the sample size), so the estimation will require additional computational resources when applied to larger regions on the genome. Second, we determined in simulations that aggregating local heritability estimates can lead to biased global heritability estimates, a phenomenon that has been observed previously[10], and may be attributable to LD leakage, i.e., non-zero correlations between SNPs on different blocks[12,34]. It is of future research interest to develop scalable algorithms to optimally approximate the LD for larger regions on the genome using our proposed Banded + LR representations. This advance will make HEELS applicable to estimate the total genetic variance of hundreds of thousands of markers.

## Methods
### Statistical model

Let **y** be a length-*n* vector that denotes the phenotypes of *n* samples. Denote by $\mathbf{X} \in \mathbb{R}^{n \times p}$ the genotype matrix of *n* individuals based on *p* markers or SNPs. We standardize **X** and **y** such that the variance of the phenotype is 1 and the variance of each marker-specific genotype vector is $1/p$, or $diag(\mathbf{X}^\top\mathbf{X}/n) = 1/p$. Let **S** and **R** denote the the marginal association statistics and the in-sample LD matrix, i.e., $\mathbf{S} = \mathbf{X}^\top\mathbf{y}$ and $\mathbf{R} = \mathbf{X}^\top\mathbf{X}$. (For the sake of simplicity of exposition, we omit scaling by $1/\sqrt{n}$ for **S** or $1/n$ for **R**, as the scaling does not affect the derivation of the HEELS estimator.) Our goal is to develop a heritability estimator using the two statistics (**S**, **R**), which attains comparable statistical efficiency as the REML estimator based on individual-level data (**X**, **y**). We start by considering the likelihood function, assuming individual-level data can be accessed.

We use the standard genetic association model with additive effects, $\mathbf{y} = \mathbf{X}\boldsymbol{\beta} + \epsilon$, where $\boldsymbol{\beta}$ is a $p \times 1$ vector assumed to follow $N(0, \sigma_g^2\mathbf{I}_p)$, and $\epsilon$ is a length-*n* vector distributed as $\epsilon \sim N(0, \sigma_e^2\mathbf{I}_n)$. Under these assumptions, we have $\mathbf{y} \sim N(0, \mathbf{V})$, where $\mathbf{V} \equiv var(\mathbf{y}) = \sigma_g^2\mathbf{X}\mathbf{X}^\top + \sigma_e^2\mathbf{I}_n$. We define SNP-heritability as the following,

$$h_{SNP}^2 := \frac{Var(\mathbf{X}\boldsymbol{\beta}|\mathbf{X})}{Var(\mathbf{y}|\mathbf{X})} = \frac{Var(\mathbf{X}\boldsymbol{\beta}|\mathbf{X})}{Var(\mathbf{X}\boldsymbol{\beta}|\mathbf{X}) + \sigma_e^2} = \frac{tr(\sigma_g^2\mathbf{I}_p\mathbf{X}^\top\mathbf{X})/n}{tr(\sigma_g^2\mathbf{I}_p\mathbf{X}^\top\mathbf{X})/n + \sigma_e^2} = \frac{\sigma_g^2}{\sigma_g^2 + \sigma_e^2}.$$

The log-likelihood function for $(\sigma_g^2, \sigma_e^2)$ is,

$$\ell(\mathbf{y}; \sigma_g^2, \sigma_e^2) = -\frac{1}{2}ln|\mathbf{V}| - \frac{1}{2}\mathbf{y}^\top\mathbf{V}^{-1}\mathbf{y}. \qquad (5)$$

Henderson developed a set of equations[16,22], known as the mixed model equations (MME), which maximize the joint density of the outcomes and the random effects,

$$\ell(\mathbf{y},\boldsymbol{\beta}; \sigma_g^2, \sigma_e^2) = -\frac{1}{2\sigma_e^2}(\mathbf{y} - \mathbf{X}\boldsymbol{\beta})^\top(\mathbf{y} - \mathbf{X}\boldsymbol{\beta}) - \frac{1}{2\sigma_g^2}\boldsymbol{\beta}^\top\boldsymbol{\beta}$$
$$- \frac{n}{2}log(\sigma_e^2) - \frac{p}{2}log(\sigma_g^2). \tag{6}$$

The Best Linear Unbiased Predictor (BLUP), which are estimates for the random effects from these MMEs, can be plugged into the score equations for the likelihood in (5), which generates an iterative procedure for estimating the variance components.[17,21] We exploit the "dual" form of this algorithm, which gives rise to the HEELS estimator. For simplicity, we have assumed that all of the observable environmental factors have been projected out, but the covariates (i.e. fixed effects) can be easily incorporated back into the model by adopting the restricted maximum likelihood approach using the projection matrix[75,76].

### The HEELS procedure

HEELS uses the marginal association statistics $\mathbf{S} = \mathbf{X}^\top\mathbf{y}$ and the empirical LD matrix $\mathbf{R} = \mathbf{X}^\top\mathbf{X}$ to solve for the variance component estimates that maximize the likelihood in Eq. (5), alternating between updating the BLUP estimates $\hat{\boldsymbol{\beta}}$ and updating the variance component estimates $(\hat{\sigma}_g^2, \hat{\sigma}_e^2)$ until convergence. The marginal likelihood in Eq. (5) can be expressed using the joint likelihood Eq. (6) and the probability of the causal effects, using the partition theorem (Supplementary Notes),

$$\ell_{HEELS}(\mathbf{S}, \mathbf{R}; \sigma_g^2, \sigma_e^2) = -\frac{1}{2}log|\sigma_e^2\mathbf{I}_n| - \frac{1}{2}log\left|\mathbf{I}_p + \frac{\sigma_g^2}{\sigma_e^2}\mathbf{R}\right|$$
$$- \frac{1}{2\sigma_e^2}\left(\mathbf{y}^\top\mathbf{y} - \mathbf{S}^\top\left(\frac{\sigma_e^2}{\sigma_g^2}\mathbf{I}_p + \mathbf{R}\right)^{-1}\mathbf{S}\right). \tag{7}$$

It can be shown that the score equations of $(\sigma_g^2, \sigma_e^2)$ derived from the likelihood in Eq. (5) are identical to the score equations based on Eq. (7), giving rise to our HEELS updating Eqs. (2) and (3) (Supplementary Notes). The BLUP estimates can be viewed as ridge estimators of the joint effect sizes, where the penalty coefficient is set to the current value of $\sigma_e^2/\sigma_g^2$ at each iteration.

We emphasize that the HEELS algorithm does not entail any standalone individual-level data $\mathbf{X}$ or $\mathbf{y}$, but only depend on the summary statistics $\mathbf{S}, \mathbf{R}$. The involvement of the variance term for phenotypes $\mathbf{y}^\top\mathbf{y}$ in Eq. (3) does not necessarily imply a requirement to access $\mathbf{y}$, especially if the phenotypes and genotypes are standardized (see similar discussions in Zou et al.[20]). In practice, when only the Z-statistics are available, we scale the estimates of $\sigma_g^2, \sigma_e^2$ at each iteration as: $[\sigma_g^2]^{(t+1)} = [\sigma_g^2]^{(t)}/([\sigma_g^2]^{(t)} + [\sigma_e^2]^{(t)}), [\sigma_e^2]^{(t)} = [\sigma_e^2]^{(t)}/([\sigma_g^2]^{(t)} + [\sigma_e^2]^{(t)})$, which is equivalent to approximating $\mathbf{y}^\top\mathbf{y}/n$ by 1. We conducted extensive simulations to validate that the HEELS estimator remains unbiased when such an approximation is applied, although the efficiency of the estimator can be slightly affected. We note that the HEELS procedure can be viewed as equivalent to an EM algorithm (Supplementary Notes), where the resulting estimates are stable numerically and belong to the parameter space[77]. This is an important feature that is not universally shared with other REML estimation algorithms such as the Anderson's algorithm[21,78] and the Newton-Raphson's algorithm[79].

### Standard error of the HEELS estimator

We derive the analytic variance of the HEELS estimator using the Fisher information matrix. For true values of the variance components $\sigma_e^2, \sigma_g^2$ and a positive definite variance-covariance matrix $\mathbf{V}$, the Fisher

Information Matrix is (Supplementary Notes):

$$I(\sigma_e^2, \sigma_g^2; \mathbf{X}, \mathbf{y}) = \frac{1}{2}\begin{bmatrix} tr(\mathbf{V}^{-2}) & tr(\mathbf{V}^{-1}\mathbf{X}\mathbf{X}^\top\mathbf{V}^{-1}) \\ tr(\mathbf{V}^{-1}\mathbf{X}\mathbf{X}^\top\mathbf{V}^{-1}) & tr(\mathbf{V}^{-1}\mathbf{X}\mathbf{X}^\top\mathbf{V}^{-1}\mathbf{X}\mathbf{X}^\top) \end{bmatrix}.$$

Using properties of trace and the identity of $\mathbf{W} := \frac{\sigma_e^2}{\sigma_g^2}\mathbf{I}_p + \mathbf{X}^\top\mathbf{X}$, we can rewrite the Fisher information using the summary statistics and the LD matrix as the following[77]:

$$I(\sigma_e^2, \sigma_g^2; \mathbf{S}, \mathbf{R}) = \frac{1}{2}\begin{bmatrix} \frac{n-p}{\sigma_e^4} + \frac{1}{\sigma_g^4}tr(\mathbf{W}^{-2}) & \frac{1}{\sigma_g^2}tr(\mathbf{W}^{-1}) - \frac{\sigma_e^2}{\sigma_g^6}tr(\mathbf{W}^{-2}) \\ \frac{1}{\sigma_g^4}tr(\mathbf{W}^{-1}) - \frac{\sigma_e^2}{\sigma_g^6}tr(\mathbf{W}^{-2}) & \frac{p}{\sigma_g^4} - \frac{2\sigma_e^2}{\sigma_g^6}tr(\mathbf{W}^{-1}) + \frac{\sigma_e^4}{\sigma_g^8}tr(\mathbf{W}^{-2}) \end{bmatrix}, \tag{8}$$

where $\mathbf{W}$ is defined in the same way as in Eq. (1). To obtain a variance estimator for $h_{SNP}^2 \equiv \frac{\sigma_g^2}{\sigma_g^2 + \sigma_e^2} := f(\sigma_g^2, \sigma_e^2)$, we apply the multivariate Delta method, and use the plug-in estimator:

$$\widehat{Var}(\hat{h}_{HEELS}^2) = \nabla f(\hat{\sigma}_e^2, \hat{\sigma}_g^2)^\top I(\hat{\sigma}_e^2, \hat{\sigma}_g^2; \mathbf{S}, \mathbf{R})^{-1}\nabla f(\hat{\sigma}_e^2, \hat{\sigma}_g^2), \tag{9}$$

where $\nabla f(\hat{\sigma}_e^2, \hat{\sigma}_g^2) = (-\frac{\hat{\sigma}_g^2}{(\hat{\sigma}_g^2 + \hat{\sigma}_e^2)^2}, \frac{\hat{\sigma}_e^2}{(\hat{\sigma}_g^2 + \hat{\sigma}_e^2)^2})$. Importantly, Eq. (9) only involves $\mathbf{S}, \mathbf{R}$, so we can compute the asymptotic variance of $h_{HEELS}^2$ without individual-level data.

### Low dimensional representations of the LD matrix

We adopt an optimization approach to solve for the best representation of LD with a "Banded + LR" structure, minimizing $||\mathbf{R} - \widetilde{\mathbf{R}}||_F^2$, where $\widetilde{\mathbf{R}}$ is the working approximating matrix in the form specified in the third column of Table 2. For example, the "Joint_Band_LR" approach simultaneously solves for the banded and the low-rank components of the approximation using PSD matrices,

$$\widetilde{\mathbf{L}}^b, \widetilde{\mathbf{U}}^r = \arg\min_{\mathbf{L}^b, \mathbf{U}^r \in L_p(\mathbb{R})} ||\mathbf{R} - \mathbf{L}^b\mathbf{L}^{b\top} - \mathbf{U}^r\mathbf{U}^{r\top}||_F^2 \tag{10}$$

where $L_p(\mathbb{R})$ denotes the set of $p \times p$ lower triangular matrices with real entries. The "PSD_Band_LR" strategy involves a two-step procedure,

$$\widetilde{\mathbf{L}}^b = \arg\min_{\mathbf{L}^b \in L_p(\mathbb{R})} ||\mathbf{R} \odot \mathbf{1}_b - \mathbf{L}^b\mathbf{L}^{b\top}||_F^2 \tag{11}$$

$$\widetilde{\mathbf{U}}^r = \arg\min_{\mathbf{U}^r \in L_p(\mathbb{R})} ||\mathbf{R} - \widetilde{\mathbf{L}}^b\widetilde{\mathbf{L}}^{b\top} - \mathbf{U}^r\mathbf{U}^{r\top}||_F^2 \tag{12}$$

where $\mathbf{1}_b$ denotes a square matrix with only the $b$ central band equal to 1 and the rest set to 0, and $\odot$ is the Hadamard product. In both cases, we approximate $\mathbf{R}$ as $\widetilde{\mathbf{L}}^b\widetilde{\mathbf{L}}^{b\top} + \widetilde{\mathbf{U}}^r\widetilde{\mathbf{U}}^{r\top}$.

Below we briefly explain the differences between the low-dimensional representation strategies listed in Table 2. When the banded component is a diagonal matrix, the representation is a special case of the "Banded + LR" structure, with $b = 1$, which corresponds to a spiked covariance model[35,80]. We estimate the diagonal elements of the covariance matrix either by taking the smallest eigenvalue of a sub-sample of the LD matrix[81] ("Spike_LR"), or via joint optimization which solves for the diagonal and the low-rank matrices simultaneously. In the latter case, we allow elements of the diagonal matrix to be either identical ("Spike_PSD") or different ("Spike_PSD_hetero").

The "Seq_Band_LR" strategy first bands the LD matrix and then performs low-rank decomposition on the residual off-banded matrix. It does not make any PSD assumption on the constituents of its solution. The main advantage of this strategy is that it most accurately represents the banded structure of the LD matrix, i.e., the banded component matches exactly the central band of the original LD matrix. A

drawback of this approach is its lack of flexibility in approximating the off-central-band structure.

In contrast, the "Joint_Band_LR" strategy jointly approximates the banded and low-rank components using PSD matrices. It provides greater flexibility of approximation for all elements indiscriminately, as the elements of the banded matrix and the low-rank matrix are jointly optimized to minimize the Frobenius norm of the error matrix. A benefit of "Joint_Band_LR" over "Seq_Band_LR" is that due to its explicit minimization of the approximation error, its solution is, by definition, closer to the original LD matrix. A potential shortcoming of the "Joint_Band_LR" strategy, however, is that by imposing a PSD assumption on both the banded and the low-rank components of the LD representation, we constrain the solution to a PSD matrix and it is more computationally intensive to solve.

To strike a balance between accurately representing the central band of the LD matrix and ensuring the computational efficiency of our algorithm, we developed a hybrid strategy, "PSD_Band_LR", which first approximates the banded component of the LD matrix using a PSD matrix followed by a low-rank decomposition of the residual matrix. On one hand, it differs from "Seq_Band_LR" in that the banded component of the approximation is guaranteed to be PSD. On the other hand, "PSD_Band_LR" differs from "Joint_Band_LR" in that it sequentially solves for the banded and the low-rank constituents of the representation, as opposed to simultaneously. As a result, "PSD_Band_LR" preserves the original structure of the LD matrix better than "Joint_Band_LR" does, leading to a more accurate representation of the central band. Finally, compared to "Seq_Band_LR", "PSD_Band_LR" produces LD approximations that allow for more stable HEELS estimation (Supplementary Notes).

## Comparison of different LD approximation strategies

We introduce a unified framework for contrasting the different LD approximation approaches in the context of heritability estimation. For a given LD representation—characterized by its structure ("Strategy") and the values of its hyper-parameters: the band size $b$ and the rank of the low rank component $r$. We considered two measures to evaluate its performance: (1) the LD approximation accuracy, measured by the ratio of the Frobenius-norms between the approximation matrix and the target matrix, $\frac{||\mathbf{R}_b + \mathbf{R}_r||_F}{||\mathbf{R}||_F}$, and (2) cross-validation (CV) bias in $\hat{h}_{HEELS}^2$, estimated using synthetic phenotypes generated from real in-sample genotypes. The first measure assesses the goodness of the approximation in general, irrespective of the genetic architecture of a trait, whereas the second measure evaluates the approximation performance of the LD representation in the context of heritability estimation.

For low-dimensional representations with smaller $b$ and $r$, the performance of the HEELS estimator can be greatly influenced by the assumed structure of the approximation strategy and the hyperparameter values. For example, for each of the "Banded + LR" strategies, we identified an approximate range of an underlying transition point ($b_{min}$) that marks the minimum optimal value of $b$, i.e., widening the bandwidth up to this value can substantially reduce the bias in $\hat{h}_{HEELS}^2$, but further increasing $b$ beyond this threshold value has a diminishing "de-biasing" effect on $h_{HEELS}^2$ (Supplementary Fig. 10).

Comparing across the three "Banded + LR" strategies, we we found that banding the full LD first and then approximating the residual using a low-rank matrix ("Seq_Band_LR") yields the most unbiased heritability estimates when $b$ and $r$ are sufficiently large, but can produce estimates with large bias if $b$ and $r$ are not appropriately chosen; approximating the LD matrix by simultaneously solving for the banded and low-rank components ("Joint_Band_LR") leads to less-biased estimates even when $b$ and $r$ are sub-optimal, although solving for the approximation is more computationally expensive than "Seq_Band_LR"; the hybrid strategy which approximates the banded

component using a PSD matrix first and finds the low-rank decomposition of the residual ("PSD_Band_LR") is least sensitive to changes in hyperparameter values and produces heritability estimates that are most stable across LD approximation settings.

## Algorithms for hyperparameter tuning

An important aspect of our LD approximation algorithm is hyperparameter tuning. While heuristics or prior knowledge about the structure of the LD may be used to determine the optimal values of ($b$, $r$), we used a more principled way to evaluate the performance of low-dimensional LD representations. We propose using a data-adaptive procedure to identify the best low-dimensional representation of the LD matrix, using simulated phenotypic data and cross-validation (Supplementary Notes). To facilitate searching of the optimal number of low-rank factors, $r^*$, we implemented our own version of the incremental SVD algorithm, where the number of low-rank factors increases step by step and the performance of the HEELS estimator is evaluated dynamically. To speed up the low-rank decomposition, we replaced the exact solutions based on direct eigen-decomposition with the approximate solutions, and compared two approaches. One is the optimization approach ("optim"), where we solve $\arg\min_{\mathbf{U}^r \in \mathbb{K}^{p \times p}} ||\mathbf{R}_{resid} - \mathbf{U}^{r\top}\mathbf{U}^r||_F^2$. $\mathbf{R}_{resid}$ is the residual off-banded component to be decomposed or approximated, and $\mathbb{K}^{p \times p}$ is the collection of lower triangular matrices. The other approach is based on randomization or sketching ("random"), where we applied fast PCA to decompose $\mathbf{R}_{resid}$ via SVD[81]. The simulation results indicate that using the optimization approach leads to better approximation of the LD and less variable $h_{SNP}^2$ estimates (Supplementary Fig. 14), though the random SVD approach may become advantageous in applications with larger LD size.

## Simulation framework

We obtained 332, 340 unrelated White British individuals by extracting samples with self-reported British ancestry who are more than third-degree relatives and excluding subjects with putative sex chromosome aneuploidy. We used genotyped array SNPs only to ensure high quality of measurement, and filtered out variants with genotype missingness > 0.01 or have minor allele frequency (MAF) <0.01. For experiments that involve subsets of the UKB individuals, we recalculated MAF and re-applied the filters above to the subsample. The environmental effects are drawn independently from the genetic effects for each individual. This ensures that population structure or cryptic relatedness among individuals have minimal impact on our estimates in the simulations.

We simulated phenotypes with different genetic architectures using real genotypes from the UK Biobank, varying sample size ($n$), the ratio of $n$ to the number of markers ($p$) and the degree of polygenicity ($p_{causal}/p$). Given the raw genotype matrix $\mathbf{G}$, we first standardized the genotype matrix: for each SNP $j$ and individual $i$, we generated $\mathbf{X}_{ij} = (\mathbf{G}_{ij} - 2f_j)/\sqrt{2f_j(1 - f_j)}$, where $\mathbf{G}_{ij} \in \{0, 1, 2\}$ and $f_j$ is the in-sample MAF of SNP $j$. For a given degree of polygenicity, we randomly sampled $p_{causal}$ markers. Denote the set of causal markers by $C$. We drew standardized effect sizes from the distribution, $\beta_j \sim N(0, \sigma_g^2/p_{causal}), \forall j \in C$, and simulated the phenotype of the $i$-th individual using $y_i = \sum_{j \in C} \mathbf{X}_{ij}\beta_j + \epsilon_i$, where $\epsilon \sim N(0, (1 - \sigma_g^2)\mathbf{I}_n)$. For each genetic architecture, we generated 100 phenotype replicates and obtained 100 estimates using each of the methods we included in the benchmark. Given phenotypes $\mathbf{y} = (y_1, ..., y_n)^\top$ and genotypes $\mathbf{X} = (\mathbf{X}_{\cdot 1}, ..., \mathbf{X}_{\cdot p})$, we computed the marginal association statistics using OLS: $\hat{\beta}_j = \mathbf{X}_j^\top \mathbf{y}/n$. Unless otherwise specified, we used the exact in-sample LD without approximation in the HEELS estimation.

We applied two software to compute the REML estimates. We used GCTA when sample size is small and used BOLT-REML when $n$ exceeds 30,000. Our primary metric of interest is relative efficiency, which is defined as the ratio of the variance of the REML estimator and the variance of a given heritability estimator. We also compared the

bias and the mean squared error of the estimators across different simulation settings.

## UKB empirical analysis

We applied HEELS to analyze the local heritability of various traits using the variant-level statistics from association studies of unrelated white British individuals in the UK Biobank, using common variants with MAF >1% on the UK Biobank Axiom array ($n = 332{,}430$, $p = 533{,}169$). We estimated the local SNP-heritability of 30 phenotypes from different trait domains, including anthropometric traits (e.g., height, BMI), hematological traits (e.g., mean corpuscular volume, red blood cell count, white blood cell count, platelet count), lipid or metabolic traits (e.g., LDL, HDL), lung function traits (e.g., forced expiratory volume, forced vital capacity, peak expiratory flow rate), behavioral traits (e.g., smoking behavior, educational attainment, neuroticism) and immunologically relevant traits (e.g., autoimmune conditions, asthma, eczema and dermatologic diseases).

We used PLINK to exclude SNPs with MAF <1%, genotype missingness >1%, genotyping rates <90%, and Hardy-Weinberg disequilibrium $p$-value <1e-6. We generated the genome-wide association statistics using the linear association analyses in PLINK, controlling for age, sex, and the top 40 genetic principal components provided by the UKB[82]. We used BOLT-REML to compute the REML estimates in the UK Biobank, and used the HESS software to compute both the HESS estimates (default setting) and the GRE estimates (i.e., no regularization of the in-sample LD matrix), after verifying that the estimates from the two methods indeed coincide when the number of eigenvectors used in TSVD equals to the rank of the genotype matrix. We used the 1,703 approximately independent loci estimated by Berisa and Pickrell[37] to define the regions as units of analysis (1.6 Mb on average or 300-400 markers per block using the genotype array data), since these non-overlapping LD blocks been been widely used in previous studies[38–40] and have been demonstrated to capture the true correlation structures among genotypes reasonably well[83].

## URLs

The following software packages were used in simulation studies and real data analyses: GCTA (https://yanglab.westlake.edu.cn/software/gcta/#Download); LD Score regression (https://github.com/bulik/ldsc); GRE (https://github.com/bogdanlab/h2-GRE); HESS (https://github.com/huwenboshi/hess); BOLT-REML v2.3.4 (https://data.broadinstitute.org/alkesgroup/BOLT-LMM/). We also used Python (3.6.3) to perform statistical analyses and used R (3.6.3) for data visualization.

## Reporting summary

Further information on research design is available in the Nature Portfolio Reporting Summary linked to this article.

## Data availability

The individual-level genotypic and phenotypic data from the UK Biobank are available under restricted access (https://www.ukbiobank.ac.uk). Our access to the individual-level data was approved under application number 52008. All data supporting the findings in our manuscript are described in the articl and the supplementary information files, or from the corresponding author upon request. The testing data which consists of the empirical LD matrices of chromosome 22 derived from the genotype array data of the UK biobank and the simulated phenotypes have been deposited on Zenodo under the link: https://zenodo.org/records/7618667.

## Code availability

Our method (HEELS) has been implemented as an open-source Python package, available on Github at https://github.com/huilisabrina/HEELS[84].

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

## Acknowledgements

We are very grateful to Luke O'Connor, Huwenbo Shi, Alkes Price, Samuel Kou, Ben Neale and Shamil Sunyaev for their helpful discussions and feedback. We thank the participants of the individual in the UK Biobank. This work was supported by funding support from the Office of Naval Research ONR-N000142112841 (to R.M.), and grants R35-CA197449, U19-CA203654, R01-HL163560, U01-HG012064, and U01-HG009088 (to X.L.).

## Author contributions

H.L., R.M. and X.L. conceived and designed the experiments. H.L. performed the experiments and the statistical analyses. H.L. wrote the manuscript with the participation of R.M. and X.L.

## Competing interests

X.L. is a consultant of AbbVie Pharmaceuticals and Verily Life Sciences. The remaining authors declare no competing interests.
