## [Peer Review File · Nature Communications]

Accurate and Efficient Estimation of Local Heritability using Summary Statistics and the Linkage Disequilibrium MatrixREVIEWER COMMENTS

Reviewer #1 (Remarks to the Author):

Overview: Li et al present HEELS, an approach to estimate SNP heritability using summary statistics and reference LD information. HEELS builds on previous efforts estimating SNP heritability using summary data (e.g., HESS, LDSC, GRE), however rather than use an empirical moment-based solution, it relies on MLE/REML estimates using a score-based approach. Similarly, the manuscript describes a novel approach to represent LD information resulting in large storage savings. The authors perform simulations and real-data analyses to demonstrate the improved statistical efficiency of HEELS over moment-based estimators. I found the manuscript to be exceptionally well written, with helpful balance between details and results split between primary/supplementary info. I have only a few comments.

Major Comments:

1. While the authors are very clear the method assumes only a single genetic variance component (and that extensions are left for future work), it would be helpful to see the performance of HEELS under simulations where MAF, LD, and functional annotations play a role in shaping genetic architecture, and how much information loss there is when inferring a single aggregate parameter.
2. The authors state the improved computational performance of their approach by using banded/low-rank LD architectures. I think it is fair to describe only the cost/runtime of downstream inference (which is similar to runtime of GCTA, not accounting for GRM estimation cost), but it would still be informative to give a sense on the computational time of LD approximation + inference.

Minor Comments:

1. Line 91 defines S as the marginal regression of X onto y divided by \sqrt{N} , but line 97 redefines S as marginal regression of X onto y . [Similarly for R with and without dividing by n ; and again in “Statistical Model” section for Online Methods]
2. Lines 184/5: “no previous study has systematically compared ...” I recommend citing Benner et al AJHG 2017 (doi.org/10.1016/j.ajhg.2017.08.012), which investigated impact of estimated LD [both empirical and shrinkage estimators] on fine-mapping.
3. Figure 2: “green – 95%” should be “blue”?

Reviewer #2 (Remarks to the Author):

In this paper, the authors proposed a new estimator, “HEELS”, for local heritability. The proposed method attains comparable statistical efficiency as REML by only using summary statistics. Moreover, the authors also proposed a unified framework to approximate the empirical LD matrix by decomposing it into the sum of a banded matrix and a low-rank matrix. Such an approximation can not only reduce the storage and memory cost of using the LD matrix, but also improves the computation efficiency of the proposed HEELS. Overall, the methodology is sound and the paper is well-written. Below are my comments and concerns on the paper.

- On page 9 lines 232-233, the authors mentioned that for fine-tuned approximation settings, the bias in the heritability estimates is even smaller than that when the full LD matrix is used. Are there any explanations why using the approximated LD matrix can have superior performance in terms of bias than using the exact LD matrix?
- The number of SNPs p in Figure 1 is 9,205 and the number of SNPs p in Figure 2 and Table 3 is both 9,220. Since the SNPs in all simulations are from chromosome 22, any explanations for why fewer SNPs were used in the simulations for Figure 1?
- All simulations conducted are based on the low-dimensional setting, i.e., the sample size is larger than the number of SNPs. What is the performance of HEELS under the high-dimensional setting?
- On page 15 and the corresponding part in the supplemental notes, the formula for h_{SNP}^2 is defined as $\text{Var}(X_i' \beta | X) / \text{Var}(y_i | X)$. What are X_i and y_i ? Are they the SNPs and the phenotype for the i th individual, respectively? If so, then the heritability depends on a specific sample, which does not make sense. But based on the formula involving the trace, I am guessing it should be the sum across all individuals in both the numerator and the denominator. Some modifications need to be made here.
- Multiple notation inconsistencies appear in the manuscript, which may cause confusion in reading. Here are some that I found:
 1. On page 16 and the corresponding part in the supplemental notes, when discussing the HEELS procedure, the marginal association statistics and the LD matrix are defined without dividing by \sqrt{n} and n , respectively. These are inconsistent with the ones defined on the previous page.
 2. The subscripts and superscripts are used interchangeably in 4 HEELS estimation with unknown sample variance on page 9 of the supplemental notes. The authors mentioned that superscripts are used to

distinguish the models for different markers. However, in the formula $y = X_j \beta_j + e_j$, the subscript was used for the j th column in the SNP matrix X . Such an issue occurred more frequently in the derivation of y'/n . In the line just above line 159, the σ_e^j and SE^j should be defined as well.

3. On page 13 in the supplemental notes, how to get from $n\beta_J' R \beta_J$ to the numerator in the next equation is not clear to me. On page 12, β_M is defined as $R \beta_J$. If so, why is there an additional n in the next equation? Maybe it is due to the inconsistency of the notation of the LD matrix mentioned above. Although the authors mentioned $\beta_J = \Sigma^{-1} \beta_M$, Σ is not defined anywhere else. More explanations are needed here.

- There are some typos in the manuscript and in the supplemental notes. Here are some that I found.

1. On page 21 line 514, "...out variants with genotype missingness > 0.01 or have MAF greater than 0.01." Since the main text is talking about common variants, should it be filtering out genotypes having MAF less than 0.01?

2. On page 21 line 528 and line 529, the meanings of x_j and X_j should be made clearer. They are not the same based on my understanding. If X_j is the j th column of the matrix X , j should be in the superscript position.

3. On page 8 in the supplemental notes, the numerator in the formula for σ_e^2 should be $y'(y - X \beta)$

4. On page 12 in the supplemental notes, "..., where c signifies chromosom," there is an 'e' missing in the chromosome.

Response to Reviewer #1:

Overview: Li et al present HEELS, an approach to estimate SNP heritability using summary statistics and reference LD information. HEELS builds on previous efforts estimating SNP heritability using summary data (e.g., HESS, LDSC, GRE), however rather than use an empirical moment-based solution, it relies on MLE/REML estimates using a score-based approach. Similarly, the manuscript describes a novel approach to represent LD information resulting in large storage savings. The authors perform simulations and real-data analyses to demonstrate the improved statistical efficiency of HEELS over moment-based estimators. I found the manuscript to be exceptionally well written, with helpful balance between details and results split between primary/supplementary info. I have only a few comments.

Major Comments:

1. While the authors are very clear the method assumes only a single genetic variance component (and that extensions are left for future work), it would be helpful to see the performance of HEELS under simulations where MAF, LD, and functional annotations play a role in shaping genetic architecture, and how much information loss there is when inferring a single aggregate parameter.

Response: Thank you very much for your comment. We included additional simulation results to showcase the performance of HEELS compared to GRE and LDSC, when the true genetic architecture is mis-specified. Specifically, for MAF-dependent architecture, we incorporated an α factor to control the strength of MAF dependency and model the SNP-specific genetic variance σ_j^2 as a function of its allele frequency f_j , $\sigma_j^2 \propto (f_j(1 - f_j))^{1+\alpha}$ (Schoech *et al.* 2019); for LD-dependent architecture, we simulated the effect sizes such that a variant's genetic variance is inversely proportional to its level of LD (Speed *et al.* 2019). We further considered scenarios where 90% or 99% of the markers have null effects.

We observed that consistent with the existing findings of the GREML performance, HEELS is not robust to these mis-specified models and can produce biased heritability estimates (Response Figure 1). These results are not surprising, but instead are aligned with our expectations, given that HEELS behaves almost identically as individual-data based GREML, which has been shown to be biased under mis-specified genetic architectures (Hou *et al.* 2019, Evans *et al.* 2018). Note that our simulations relating to the sparse architecture (*i.e.* two-component mixture) can be viewed as incorporating a simple binary annotation in the generative model, but we defer a more detailed analysis of how functional annotations affect our estimator to future studies. Notably, the bias of the HEELS estimator is smaller than that from the LDSC, and the MSE of HEELS is the lowest among the summary-statistics-based methods (Response Figure 2). We have discussed in the Discussion section the future research direction of extending HEELS to allow for the dependence of heritability on annotations.

We have discussed these results in Line 344-349 of the main text and Supplementary Figure 20-21.

Response Figure 1. Distribution of the heritability estimates from HEELS under mis-specified models. The simulation results are based on real genotypic data from random subsets of unrelated individuals in the UK Biobank, array SNPs on chromosome 22 with MAF > 0.01. The α value determines the strength of the MAF-dependency, with $\sigma_j^2 \propto (f_j(1 - f_j))^{1+\alpha}$. Setting α to -1 corresponds to the GCTA model; setting α to -0.25 corresponds to the LDAK model. LD-dependent architecture: assume the genetic variance of a variant is inversely proportional to its level of LD. Red-dotted line: true heritability of 0.25. The row panels represent polygenicity or the proportion of SNPs with non-zero effects.

Response Figure 2. Comparison of MSE between different summary-statistics-based heritability estimation methods (HEELS, GRE and LDSC) under mis-specified models. The simulation results are based on real genotypic data from random subsets of unrelated individuals in the UK Biobank, array SNPs on chromosome 22 with MAF > 0.01. The α value determines the strength of the MAF-dependency, with $\sigma_j^2 \propto (f_j(1 - f_j))^{1+\alpha}$. Setting α to -1 corresponds to the GCTA model; setting α to -0.25 corresponds to the LDAK model. LD-dependent architecture: assume the genetic variance of a variant is inversely proportional to its level of LD. Red-dotted line: true heritability of 0.25. The row panels represent polygenicity or the proportion of SNPs with non-zero effects, e.g., 0.1 means 10% of the SNPs are causal.

2. The authors state the improved computational performance of their approach by using banded/low-rank LD architectures. I think it is fair to describe only the cost/runtime of downstream inference (which is similar to runtime of GCTA, not accounting for GRM estimation cost), but it would still be informative to give a sense on the computational time of LD approximation + inference.

Response: Thank you very much for your comment. We now report the computational runtime cost for all of the different LD approximation methods we have compared (*i.e.*, all settings that are included in the benchmarking in our main text Figure 2). Compared to other approximation methods, which require banding only, low-rank decomposition only or assume a spiked covariance structure, our LD decompositions take longer to compute. Nevertheless, we note that this is a one-time cost, and such banded + low-rank approximation can greatly benefit the HEELS estimation by improving the computational efficiency of the downstream analyses, without incurring large loss of accuracy or efficiency.

Please review Supplementary Tables 4-5 which we newly added. We updated sections of the that reference these statistics as well, for example, in the caption of Table 3.

Minor Comments:

1. Line 91 defines S as the marginal regression of X onto y divided by \sqrt{N} , but line 97 redefines S as marginal regression of X onto y . [Similarly for R with and without dividing by n ; and again in “Statistical Model” section for Online Methods]

Response: Thank you very much for pointing it out. We have modified the notation to make them consistent across the text. Since we assume that the estimated LD matrix is in-sample, the sample size used for the association statistics and for LD is identical. As a result, our derivation of the HEELS estimator is not affected by the scaling, by the square root of N (for S) or N (for R). We adopt the notations omitting the scaling for simplicity of exposition and provide a note about it.

Please review our changes in Line 96-99, 378-381 of the main text and other related sections in the Supplementary Notes.

2. Lines 184/5: “no previous study has systematically compared ...” I recommend citing Benner et al AJHG 2017 (doi.org/10.1016/j.ajhg.2017.08.012), which investigated impact of estimated LD [both empirical and shrinkage estimators] on fine-mapping.

Response: Thank you very much for mentioning this nicely written article. We have included this reference in our discussion of the existing LD approximation methods. Benner et al. proposed a thresholding method to regularize the LD, which is different from our proposed banded + low-rank approximations. Here we make note of additional differences between this work and ours.

- First, Benner *et al.* 2017 focuses on comparing the impact of different LD modeling on fine-mapping whereas our end goal is to ensure the consistency and efficiency of the heritability estimator. Notably, the primary metric evaluated in Benner et al. is the size and the coverage of credible sets of the causal variants. In contrast, we compare the performance of different LD approximation strategies using 1) the accuracy of approximation, defined as the ratio of the norm between the approximated LD matrix and

the original empirical LD matrix, and 2) the statistical property of the heritability estimator when the LD is approximated.

- Second, we emphasize in our work that the positive semi-definite (PSD) property of the approximated LD has important implications for the HEELS heritability estimator. LDstore does not regard the PSD of the LD approximation as necessary in the shrinkage procedure, and does not consider the factor of PSD in their comparisons.

Please review our changes in Line 185-186 of the main text.

3. Figure 2: “green – 95%” should be “blue”?

Thank you very much for pointing it out. We have changed the caption.

Response to Reviewer #2:

In this paper, the authors proposed a new estimator, “HEELS”, for local heritability. The proposed method attains comparable statistical efficiency as REML by only using summary statistics. Moreover, the authors also proposed a unified framework to approximate the empirical LD matrix by decomposing it into the sum of a banded matrix and a low-rank matrix. Such an approximation can not only reduce the storage and memory cost of using the LD matrix, but also improves the computation efficiency of the proposed HEELS. Overall, the methodology is sound and the paper is well-written.

Response: Thank you for your positive comments on our paper.

Below are my comments and concerns on the paper.

- On page 9 lines 232-233, the authors mentioned that for fine-tuned approximation settings, the bias in the heritability estimates is even smaller than that when the full LD matrix is used. Are there any explanations why using the approximated LD matrix can have superior performance in terms of bias than using the exact LD matrix?

Response: Thank you very much for your comment. The improved accuracy of the heritability estimator under certain settings when LD is approximated may be attributable to *regularization*, where the approximation leads to an estimate of the LD that is closer to the true covariance. Intuitively, the regularized estimators will perform better if the true covariance matrix has structures that are well-captured by constraints or assumptions imposed by the regularization, such as sparsity or banded-ness. Seminal works on covariance matrix estimation have provided high-probability analyses that bound the errors of regularized estimators when the true covariance is sparse or banded (Section 6.5 of Wainwright 2019, Cai et al. 2010, Bien et al. 2016).

In our case, we hypothesize that the LD approximation strategies that lead to more accurate heritability estimates might have better matched the original LD (in particular, the specific banded + low-rank structure assumed). In a recently published paper (Salehi et al. 2022), researchers have found that LD regularization can lead to less biased BLUP estimates. Although this paper examines the impact of the regularized LD precision matrix on the SNP-effect predictors, the better performance of the regularized LD precision matrix is consistent with our observation of lower bias in the heritability estimates in the presence of regularization.

- The number of SNPs p in Figure 1 is 9,205 and the number of SNPs p in Figure 2 and Table 3 is both 9,220. Since the SNPs in all simulations are from chromosome 22, any explanations for why fewer SNPs were used in the simulations for Figure 1?

Response: Thank you for pointing it out. The number of SNPs is different between these two sets of analyses because we applied the $MAF > 0.01$ filter separately to two different samples:

- The results in Figure 1 are generated based on a random subsample of the full UKB sample ($n=30,000$). We restricted analyses to the smaller sample due to the limited scalability of GCTA when applied to biobank sized datasets.
- On the other hand, the results in Figure 2 and Table 3 are based on using the full UKB sample ($n=332,430$).

There was no difference in the QC procedure or steps we undertook to prepare the samples for these analyses. We really appreciate your careful reading.

- All simulations conducted are based on the low-dimensional setting, i.e., the sample size is larger than the number of SNPs. What is the performance of HEELS under the high-dimensional setting?

Response: Thank you very much for your comment. We conducted additional simulation studies to investigate how HEELS performs under high-dimensional settings (Response Figure 3). In particular, we varied the ratio of sample size (n) and the effective number of variants (p). Our simulation results indicate that when sample size is much lower than the number of SNPs ($n \ll p$), the HEELS heritability estimator has some finite sample bias and a large variance, and that a lower n/p ratio produces less efficient estimator. Our empirical findings are consistent with Jiang et al. 2016 (Annal of Statistics). They showed the REML estimator is consistent when n/p goes to a constant bounded away from zero and one asymptotically. However, when n/p goes to 0 asymptotically, the REML estimator is asymptotically unbiased but its variance does not go to zero asymptotically, i.e., the estimator is inconsistent.

Response Figure 3. Heritability estimates from different simulation settings with varying sample size (n), effective number of variants (p) and true heritability. We simulated the genotypes assuming no LD correlations, so the number of markers is the effective number of SNPs (i.e. independent). Boxplots show the distribution of heritability estimates from 100 simulations. Red reference line represents the true heritability value of 0.2 (low) or 0.6 (high). The lower and upper hinges correspond to the first and third quartiles. The upper (lower) whisker extends from the hinge to the largest (smallest) value no further than $1.5 \times \text{IQR}$ from the hinge. Data beyond the end of the whiskers are outliers.

• On page 15 and the corresponding part in the supplemental notes, the formula for h_{SNP}^2 is defined as $\text{Var}(X_i'\beta|X)/\text{Var}(y_i|X)$. What are X_i and y_i ? Are they the SNPs and the phenotype for the i th individual, respectively? If so, then the heritability depends on a specific sample, which does not make sense. But based on the formula involving the trace, I am guessing it should be the sum across all individuals in both the numerator and the denominator. Some modifications need to be made here.

Response: Thank you very much for your comment. You are correct that X_i and y_i refer to the genotype vectors and the phenotype for individual i . We intended to denote the conditional variance of genetic effects and phenotypes *across the subjects* by $\text{Var}(X_i'\beta|X)$ and $\text{Var}(y_i|X)$, which are constant and do not depend on i . We have modified the notation to make it clearer.

Please review our changes to the notations used on page 15 and the corresponding part in the supplementary notes.

• Multiple notation inconsistencies appear in the manuscript, which may cause confusion in reading. Here are some that I found:

1. On page 16 and the corresponding part in the supplemental notes, when discussing the HEELS procedure, the marginal association statistics and the LD matrix are defined without dividing by root n and n , respectively. These are inconsistent with the ones defined on the previous page.

Response: Thank you very much for pointing it out. We have modified the notations to make them consistent across the texts. Since we assume that the estimated LD matrix is in-sample, the sample size used for the association statistics and for LD is identical. As a result, our derivation of the HEELS estimator is not affected by the scaling, by the square root of N (for S) or N (for R). We adopt the notations omitting the scaling for simplicity of exposition and provide a note about it.

Please review our changes in Line 96-99, 378-381 of the main text and other related sections in the Supplementary Notes.

2. The subscripts and superscripts are used interchangeably in 4 HEELS estimation with unknown sample variance on page 9 of the supplemental notes. The authors mentioned that superscripts are used to distinguish the models for different markers. However, in the formula $y = X_j \beta^j + e^j$, the subscript was used for the j th column in the SNP matrix X . Such an issue occurred more frequently in the derivation of $y'y/n$. In the line just above line 159, the σ_e^j and SE^j should be defined as well.

Response: Thank you very much for pointing it out. We have modified the notations to make them clearer. In particular, we now use β_M^j and e_M^j for effect size and error term in the marginal model, to be distinguished from the effect size and error term in the joint model.

Please review our changes to Section 4 of the Supplementary Notes.

3. On page 13 in the supplemental notes, how to get from $n\beta_J R\beta_J$ to the numerator in the next equation is not clear to me. On page 12, β_M is defined as $R\beta_J$. If so, why is there an additional n in the next equation? Maybe it is due to the inconsistency of the notation of the LD matrix mentioned above. Although the authors mentioned $\beta_J = \Sigma^{-1}\beta_M$, Σ is not defined anywhere else. More explanations are needed here.

Response: We apologize for the inconsistent notations. We have addressed your comment for the clarification of scaling by n .

We have made quite extensive changes to this section to clarify the usage of our notations. In particular, we now adopt the same notations used by the GRE and HESS paper – using Σ to denote the in-sample LD that includes the scaling of $X'X$ by n . Furthermore, use the *unstandardized* marginal effect size from OLS, which involves the scaling by $1/n$, as opposed to the standardized Z-scores (“S” defined in HEELS).

Please review our changes to Section 6 of the Supplementary Notes.

• There are some typos in the manuscript and in the supplemental notes. Here are some that I found.

1. On page 21 line 514, “...out variants with genotype missingness > 0.01 or have MAF greater than 0.01 .” Since the main text is talking about common variants, should it be filtering out genotypes having MAF less than 0.01 ?

Response: Thank you very much for pointing it out. We have made the correction.

2. On page 21 line 528 and line 529, the meanings of x_j and X_j should be made clearer. They are not the same based on my understanding. If X_j is the j th column of the matrix X , j should be in the superscript position.

Response: Thank you very much for pointing it out. We have modified the notations such that X_j denotes the j -th column of the genotype matrix and is used consistently in the text.

3. On page 8 in the supplemental notes, the numerator in the formula for σ_e^2 should be $y'(y - X\beta)$

Response: Thank you very much for pointing it out. We have made the correction.

4. On page 12 in the supplemental notes, “..., where c signifies chromosom,” there is an ‘e’ missing in the chromosome.

Response: Thank you very much for pointing it out. We have made the correction.

REVIEWERS' COMMENTS

Reviewer #1 (Remarks to the Author):

The authors have addressed all of my comments. I have no additional comments at this time.

Reviewer #2 (Remarks to the Author):

The authors have successfully addressed my previous comments and I do not have any further concerns.